# The Impact of Antiviral Treatment of Hepatitis B Virus after Kidney Transplant and the Latest Insights

**DOI:** 10.3390/pathogens12020340

**Published:** 2023-02-17

**Authors:** Fabrizio Fabrizi, Maria Francesca Donato, Federica Tripodi, Anna Regalia, Pietro Lampertico, Giuseppe Castellano

**Affiliations:** 1Division of Nephrology, Dialysis, and Kidney Transplant, Foundation IRCCS Cà Granda Ospedale Maggiore Policlinico, 20122 Milano, Italy; 2Division of Gastroenterology and Hepatology, Foundation IRCCS Cà Granda Ospedale Maggiore Policlinico, 20122 Milano, Italy; 3Department of Clinical Sciences and Community Health, University of Milan, 20122 Milano, Italy

**Keywords:** hepatitis B virus, kidney transplantation, nucleos(t)ide analogues, survival

## Abstract

Background: The current frequency of hepatitis B virus infection in patients with advanced chronic kidney disease (CKD) (including patients on maintenance dialysis and kidney transplant recipients) is low but not negligible worldwide. HBV has a deleterious effect on survival after a kidney transplant; antiviral treatments improved the short-term outcomes of kidney transplant recipients, but their long-term impact remains uncertain. Aim: The aim of this review is to assess the role of antiviral therapy for HBV in improving survival after a kidney transplant. The recent publication of large surveys has prompted us to update the available evidence on the impact of HBV on patient and graft survival after a kidney transplant. Methods: We have conducted an extensive review of the medical literature, and various research engines have been used. Results: We retrieved several studies (*n* = 11; *n* = 121,436 unique patients) and found an association between positive serologic HBsAg status and diminished patient and graft survival after a kidney transplant; the adjusted relative risk (aRR) of all-cause mortality and graft loss was 2.85 (95% CI, 2.36; 3.33, *p* < 0.0001) and 1.26 (95% CI, 1.02; 1.51, *p* < 0.0001), respectively. To our knowledge, at least six studies reported improved patient and graft survival after the adoption of antiviral therapies for HBV (this result was reported with both survival curves and multivariable regression). According to novel clinical guidelines, entecavir has been suggested as a ‘first line’ antiviral agent for the treatment of HBV after a kidney transplant. Conclusions: The recent availability of safe and effective antiviral drugs for the treatment of HBV has meant that the survival curves of HBsAg-positive patients on antiviral therapy and HBsAg-negative patients after a kidney transplant can be comparable. Antiviral therapy should be systematically proposed to HBV-positive kidney transplant recipients and candidates to avoid the deleterious hepatic and extra-hepatic effects of chronic HBV replication.

## 1. Introduction

Hepatitis B virus infection remains a major global health problem and causes both acute and chronic liver diseases. According to the WHO’s calculations, 296 million people were living with chronic HBV infection in 2019 worldwide, with 1.5 million new infections each year. The frequency of HBV infection has decreased in the developed world, but the current numbers are probably an underestimate due to the under-sampling of immigrant groups, which still have a high prevalence of HBV infection [1].

Despite advances in prevention and therapy, HBV is still considered an ‘incurable’ disease and remains a challenging issue in patients who have undergone kidney transplantation. The frequency of HBV infection in patients with advanced CKD is not negligible [2]; the EASL 2017 Clinical Practice Guidelines recommended screening for HBV markers in all dialysis and kidney transplant recipients. HBsAg-positive dialysis patients and kidney allograft recipients should receive antiviral agents (ETV or TAF) for treatment or prophylaxis, and close monitoring for HBV should be performed in HBsAg-negative/anti-HBc-positive kidney transplant recipients [3]. The effects of HBV infection on patient and graft survival after RT remain unclear in light of the recent adoption of antiviral therapies [1].

We have conducted a narrative review on HBV infection, kidney transplant, and antiviral therapies; a pooled analysis on the impact of HBV infection on patient and graft survival after a kidney transplant has also been carried out.

## 2. Information Sources and Search Strategy

### 2.1. Data Extraction

Electronic searches of the National Library of Medicine’s MEDLINE database and Current Contents and manual searches of selected specialty journals were performed to identify all of the pertinent literature. It was previously demonstrated that an electronic search alone may not be sensitive enough [4]. Various MEDLINE database engines (Ovid, PubMed, and GratefulMed) and Embase were used. Sources of grey literature were also retrieved. The following keywords were used: (‘Hepatitis B Virus’ OR ‘HBV’ OR ‘HBsAg’ OR ‘Hepatitis B’) AND (‘Chronic kidney disease’ OR ‘Kidney transplant’ OR ‘Renal transplant’ OR ‘Renal impairment’ OR ‘End-stage renal disease’) AND (‘Lamivudine’ OR ‘Adefovir’ OR ‘Tenofovir’ OR ‘Entecavir’ OR ‘Telbivudine’). We considered published articles from 1 January 1990 to 30 June 2022. The search was limited to human studies that involved individuals aged >18 years, and only English-language articles were included. Studies were compared to eliminate duplicate reports for the same patients, which included contacting investigators when necessary. Data extraction was carried out independently by two investigators (F.F. and F.T.), and a consensus was achieved for all data. Eligibility and exclusion criteria were prespecified.

### 2.2. Criteria for Inclusion

We included reports evaluating patients with advanced chronic kidney disease (stage 5 CKD) who underwent RT. Both case–control and cohort studies were considered eligible for inclusion in the analysis. To be considered for inclusion in our meta-analysis, reports had to define HBV infection by testing for HBsAg in serum. Information on HBsAg-seropositive status had to be registered at the time of enrolment. Patient outcomes collected included death, cause of death, and loss to follow-up.

### 2.3. Ineligible Studies

Studies were excluded if they reported inadequate data on treatments or measures of responses. Studies that were only published as abstracts or interim reports were excluded; letters and review articles were not considered for this analysis.

### 2.4. End-Points of Interest

The primary end-point was the adjusted relative risk (aRR) and 95% confidence interval (CI) of all-cause mortality among RT recipients who were HBsAg-positive relative to those not infected. The secondary end-point was the adjusted RR and 95% CI of all-cause graft loss among RT recipients who were HBsAg-positive relative to those not infected. The aRR was calculated using Cox proportional hazards analysis in each study. The aim of the Cox proportional hazards model was to assess the impact of HBsAg-seropositive status per se on patient and graft survival. The Cox proportional hazards analysis was adopted to estimate the independent effect of HBsAg-positive serologic status on survival after adjustment for different follow-up times and the distribution of potential confounders (i.e., age, gender, race, time on dialysis, etc.).

### 2.5. Statistical Methods

A summary estimate of the aRR of all-cause mortality in HBsAg-positive relative to HBsAg-negative patients was generated by weighting the study-specific RRs by the inverse of the variance. The overall estimates were generated using the random-effects model of Der Simonian and Laird. The heterogeneity was calculated by R*i* (i.e., the proportion of total variance due to between-study variance). The origin of heterogeneity was explored using stratified analysis; we identified subgroups of studies defined by study characteristics such as the country of origin (Europe/US), reference year, or size (single-centre/multi-centre). Sensitivity analysis using a fixed-effects model was also carried out to evaluate the consistency of results. Statistical analysis was performed with the software HEpiMA (version 2.1.3) and Rev Man (Review Manager) 5.0. A 5% significance was adopted for the alpha risk. Every estimate was given with its 95% CI.

## 3. Epidemiology of HBV Infection in Advanced CKD (Dialysis and Kidney Transplant)

Some information exists on the epidemiology of hepatitis B virus infection in patients on regular dialysis and after a kidney transplant. The most important survey was conducted by the Dialysis Outcomes and Practice Patterns Study (DOPPS), which is an international, prospective cohort study of patients on haemodialysis. A total of 82,449 adult patients were enrolled between 1996 and 2015, and the frequency of HBsAg-seropositive status was 2.3% (*n* = 1898) [5]. Data were collected from 76,689 patients on HD from the Western world (Australia, Belgium, Canada, France, Germany, the United Kingdom, the United States, Italy, Japan, New Zealand, Spain, and Sweden), six Gulf Cooperation Council Countries (Bahrain, Kuwait, Oman, Qatar, Saudi Arabia, and the United Arab Emirates), and other countries, including China, Romania, and Turkey. As listed in Table 1, evidence on the epidemiology of HBV infection among patients with advanced CKD in developing countries is less abundant compared to the Western world. The HBsAg prevalence rates appear to be greater in patients with advanced CKD on treatment in developing (up to 25%) compared with developed countries (up to 10%). Information on the rate of chronic HBsAg carriers among patients with advanced CKD from less developed regions is mostly based on small-sized single-centre surveys [6,7,8,9,10,11,12,13,14,15,16,17,18].

## 4. Natural History of HBV Infection in Advanced CKD (Dialysis and Kidney Transplant)

The natural history of hepatitis B in patients with advanced CKD has not been detailed with accuracy: patients with chronic HBsAg carriage are typically anicteric and do not commonly develop symptoms of hepatitis. We need large studies with long follow-ups to evaluate the course of HBV over time in advanced CKD, as hepatitis B typically progresses slowly. The life expectancy of patients with advanced CKD is shorter than that of the general population due to older age and multiple comorbidities [19,20].

The recognition of HBV-related liver disease on the grounds of biochemical tests is difficult, as serum aminotransferase values are lower in patients with advanced CKD than in the general population. We carried out a study on a large (*n* = 407) cohort of patients with advanced CKD (serum creatinine ≥ 2 mg/dL) in the predialysis stage; in an age-matched study, a comparison of serum aminotransferase values of patients with advanced CKD with those of healthy groups (age > 60 years) showed that AST and ALT levels were lower in advanced CKD (AST 17.4 ± 8 vs. 22.3 ± 6 (*p* = 0.0001) and ALT 16.3 ± 9 vs. 20.8 ± 10 (*p* = 0.008)) [21]. A survey performed on 727 patients receiving long-term dialysis in northern Italy showed that AST and ALT values were significantly greater in HBsAg-positive/HBV-DNA-positive than in HBsAg-negative patients on dialysis (AST, 22.8 ± 31.3 vs. 14.1 ± 9.7 IU/L (*p* = 0.0001); ALT, 25.07 ± 41.49 vs. 13.9 ± 41.59 IU/L (*p* < 0.0001)). Multivariable analysis revealed a significant and independent link between detectable HBsAg/HBV DNA in serum and AST (*p* = 0.0001) and ALT (*p* = 0.0001) values [22].

Additional factors that make it difficult to assess HBV-related liver disease in uraemia include limited information on liver histology. This is probably related to the reluctance of clinicians to perform liver biopsies in patients with advanced CKD for impaired platelet function in uraemia. Several agents are now available for the treatment of HBV in patients with intact kidneys and even in advanced CKD (dialysis and renal transplant), and this clearly hampers the possibility of performing large studies with appropriate follow-ups to assess the natural course of HBV over time.

Mortality is a reliable end-point in the assessment of the natural history of HBV. Ko and coworkers included 3968 patients receiving haemodialysis from nine selected centres in Hiroshima Prefecture (Japan) [8]. The prevalence of chronic HBsAg carriage decreased from 2.8% (before 1990) to 2.2% (from 1991 to 2001) and 1.3% (after 2002). In the subset of patients who started haemodialysis after 2002, positive HBsAg serologic status was associated with mortality (aHR = 2.38 (95% CI, 1.12–5.06, *p* = 0.024)). With regard to causes of death in all groups, heart failure was the leading cause of death among patients on haemodialysis (about 23%), followed by infectious and cerebrovascular disease. Liver cirrhosis and hepatocellular carcinoma accounted for 1% and 2%, respectively, of all causes of death in haemodialysis patients.

The most important and recent survey on how HBV infection affects outcomes after a kidney transplant was conducted in Korea [12]. A total of 3482 patients with functioning kidney grafts were included (*n* = 165 [4.5%] HBsAg-positive); the patients were followed up for 89.1 ± 54 months. The Cox proportional hazards analysis showed that the adjusted HR for mortality and graft failure of HBV-seropositive status was 2.37 (95% CI, 1.15–4.86, *p* = 0.019) and 1.38 (95% CI, 0.54–3.499, NS), respectively. Among the causes of mortality in patients with HBV, hepatic diseases (44.4%) (hepatocellular carcinoma and acute hepatic failure) were more frequent than others.

The progression of HBV-related liver disease after a transplant is commonly supported by multiple factors, such as alcohol abuse, HCV or HIV coinfection, and immunosuppression. It is clear that immunosuppressive agents have a permissive effect on the replication of HBV, and this facilitates an accelerated course of HBV-related liver damage (with its attendant complications—cirrhosis, hepatic failure, and hepatocellular carcinoma) among patients with functioning kidney allografts.

## 5. HBV Infection after Kidney Transplant: Impact on Patient/Graft Survival

A large body of data has already been collected regarding the course of HBV infection after a kidney transplant [23,24,25]. According to our electronic and manual searches, we retrieved 1322 papers; 122 of these were considered potentially relevant and were selected for full-text review. Of these, 11 papers fulfilled the inclusion criteria, and 111 were excluded. The list of the 122 references is available from the authors on request. There was 100% concordance between reviewers with respect to the final inclusion and exclusion of reports based on the predefined inclusion and exclusion criteria. A total of 11 papers (*n* = 121,436 unique kidney transplant recipients) fulfilled the inclusion criteria, and 111 were excluded. The majority of the reports included in the pooled analysis had a retrospective design and evaluated the impact of positive HBsAg serologic status on patient survival after a kidney transplant (Table 2 and Table 3) [10,12,14,26,27,28,29,30,31,32,33].

Shown in Table 2, Table 3 and Table 4 are some salient demographic characteristics of patients enrolled in the included studies. We observed an independent and significant influence of HBsAg infection on diminished survival; the summary estimate for the aRR of all-cause mortality with HBsAg across the identified reports was 2.85 (95% CI, 2.36; 3.33) (*p* < 0.0001) (Forest plot shown in Figure 1). The funnel plot suggests a publication bias (Figure 2). Tests for homogeneity of the aRR across the eleven studies resulted in Chi^2^ = 2177,6 and *I*^2^ = 99.5% (*p*-value from *Q*-test, 0.0001); that is, the homogeneity assumption was rejected. The pooled estimate for aRR (all-cause death) according to the fixed- or random-effects model is shown in Figure 3. The relationship between HBsAg-seropositive status and graft survival was addressed in 11 reports [10,12,14,26,27,29,30,32,33,34,35]. The overall estimate for the adjusted RR of all-cause graft loss was 1.26 (95% CI, 1.02; 1.51, *p* < 0.0001) (heterogeneity statistics, Chi^2^ = 22.7; *I*^2^ = 56% (*p*-value by *Q*-test = 0.01)) (Forest plot shown in Figure 4). Figure 5 shows the aRR of all-cause graft loss and 95% CI (individual studies).

The deleterious effects of chronic HBV on graft and patient survival after a kidney transplant are likely related to various factors, including posttransplant immunosuppression, which favours HBV DNA replication, as mentioned above. Accelerated atherogenesis induced by HBV has been mentioned in the adult general population [36], which could explain the link between HBV infection and the decline in kidney function that occurs both in patients with intact kidneys and in those who have undergone transplantation [36]. Additional extra-hepatic complications due to HBV after a renal transplant include de novo or recurrent glomerulonephritis or nephrotoxicity related to excessive exposure to calcineurin inhibitors.

## 6. HBV after Kidney Transplant: Oral Nucleos(t)ide Analogues

Lamivudine was the first oral nucleoside analogue (NA) adopted for the treatment of chronic HBV infection. Lamivudine proved to improve short-term outcomes of HBsAg-positive patients after a kidney transplant, but prolonged administration is biased by high rates of drug resistance, which can be as high as 60% after 5 years. According to a systematic review with a meta-analysis of clinical trials (*n* = 14; *n* = 184 unique patients), the pooled frequency of HBV DNA clearance and alanine aminotransferase normalisation was 91% (95% CI, 86%–96%) and 81% (70%–92%), respectively, after lamivudine therapy. Lamivudine resistance occurred in 18% (95% CI, 10%–37%) of patients on lamivudine treatment; resistance to lamivudine increased with the duration of lamivudine use (*r* = 0.620, *p* = 0.019) [37].

Both adefovir dipivoxil [38,39] and tenofovir [40,41] demonstrated efficacy in suppressing HBV DNA in patients with lamivudine-resistant HBV; however, these drugs are nephrotoxic, and their long-term use may impair kidney allograft survival. Lampertico and coworkers [38] reported on 11 kidney transplant recipients with lamivudine-resistant hepatitis B who were treated with add-on ADV. During a follow-up of 36 (12–48) months, nine patients (82%) achieved the clearance of HBV DNA, and the 3-year cumulative virologic response was 88%. None developed resistance to ADV, and the pattern of lamivudine resistance was unchanged. Six patients (55%) showed a decline in creatinine clearance (without damage to kidney proximal tubules) over a follow-up of 11 months, and the ADV dose was reduced with clinical benefit.

Recent clinical guidelines [42,43] suggested that entecavir (ETV), tenofovir disoproxil fumarate (TDF), and tenofovir alafenamide (TAF) be used in kidney transplant candidates and recipients for the treatment of HBV. ETV, TAF, and TDF are preferred nucleoside agents due to their high efficacy and high barrier to viral resistance. Tenofovir has been used as a rescue therapy for lamivudine resistance, but it has been associated with nephrotoxicity, in which up to 30–50% of kidney transplant recipients required the discontinuation of treatment [42,43]. In those patients who show kidney impairment posttransplant, ETV or TAF is preferred over TDF; however, data on the use of TAF are extremely limited in kidney transplant recipients. Antiviral therapy should be given indefinitely post-kidney transplant as long as the patient remains on immunosuppressive therapy; in addition, regular follow-up and monitoring for the response to antiviral therapy and continued surveillance for HCC should be conducted [43].

Entecavir shows excellent antiviral activity for both treatment-naïve and lamivudine-resistant patients. In addition, it exhibits a great resistance barrier in the treatment of naïve patients and does not cause nephrotoxicity. On the grounds of these characteristics, there are some data on its use in advanced CKD [44] or after a kidney transplant [45,46,47,48]. The largest series on this point has been reported by Yap and colleagues [48], who retrospectively reviewed a cohort (*n* = 30 patients) of RT recipients who received ETV during the years 2007–2017. Patients were categorised into two groups, treatment-naïve (*n* = 18) and lamivudine-resistant (*n* = 12) patients, and received ETV for 48.4 ± 35.2 and 66 ± 26 months, respectively. At 48 and 60 months, ETV was associated with frequencies of undetectable HBV DNA of 100% and 100%, respectively, in the first group. The frequencies of HBV DNA clearance were 91% and 91% at 48 and 60 months, respectively, in the second group. Kidney allograft function was stable during a follow-up of 63.2 ± 33.4 months for both groups. No difference in patient (87% vs. 83%, NS) survival between treatment-naïve and lamivudine-resistant patients occurred at 5 years. Four patients (33%) in the lamivudine-resistant group developed drug resistance, and two of them had persistent viral replication while on ETV (HBV DNA > 5 × 10^3^ IU/mL). Two additional kidney recipients developed virologic breakthrough at 40.3 ± 15 months, after HBV DNA suppression. All of them responded to TDF rescue therapy, and HBV DNA again became undetectable after 21.3 ± 12.2 months. Kidney allograft survival was 93% at 5 years in treatment-naïve and 100% in lamivudine-resistant patients (NS) [48].

## 7. HBV after Kidney Transplant: Therapy Effects of Antiviral Agents

According to our stratified analysis (Table 5), no meaningful difference in pooled aRR across many designs (i.e., population-based or recent studies) occurred (patient survival). Antiviral agents against HBV suppress HBV replication; the effect of viral replication on kidney transplant outcomes remains controversial. As listed in Table 3, many studies included in our review did not give information on the number of patients who underwent antiviral treatment before or after a kidney transplant. We assumed that the most recent surveys were those where the frequency of kidney transplant patients on antiviral drugs against HBV was greater. Lamivudine was licensed for therapy in 1996 [49,50], and subsequent antiviral agents have played a role in providing better outcomes in HBV-infected patients. The relationship between positive HBsAg status in serum and lower patient survival persisted even in the subset of recent studies (Table 5). On the contrary, no link occurred between HBsAg and graft survival in the subset of recent reports (Table 6). The information on the virological features of HBV was extremely limited in these surveys (Table 4), and the potential association between HBV genotypes or HBV viral load and survival was incompletely addressed (Table 5 and Table 6). To our knowledge, at least six studies [14,26,28,51,52,53] have formally evaluated the effects of antiviral treatment on HBV and reported consistent improvements in patient and graft survival with the adoption of antiviral therapies for HBV (Table 7) [26,28,51,52,53]. Unfortunately, no detailed information was given on the comorbidities of those patients who received antiviral therapies for HBV.

Reddy and coworkers [33] worked with a North American database of renal transplants (*n* = 3482) and identified three groups—anti-HCV-positive/HBsAg-negative patients (*n* = 55), anti-HCV-negative/HBsAg-negative (*n* = 3267), and anti-HCV-negative/HBsAg-positive (*n* = 160). They found that patient survival was significantly reduced in HBV-positive compared to HBV-negative patients in the past era (1987 to 1994); patient survival was not reduced in HBV-infected recipients in the recent era (2001 to 2007). These authors suggested various possibilities for the death reduction in HBsAg-positive recipients in the recent era, including demographic changes and antiviral treatment, among others.

Yu and colleagues [14] performed a population-based retrospective cohort study. They evaluated a large cohort of kidney transplant recipients (*n* = 4133), with data obtained from the National Health Insurance Research Database (NHIRD) in Taiwan. There were patients with neither HBV nor HCV infection (*n* = 3485), those with HBV infection (*n* = 336), those with HCV infection (*n* = 262), and those with HBV/HCV coinfection (*n* = 50). According to Yu and coworkers, lamivudine therapy significantly improved survival after a kidney transplant [14]; of note, such an association was observed by multivariate analysis. The aRR of death was 0.48 (95% CI, 0.29–0.79) among RT patients who received lamivudine compared with those who did not (*p* < 0.01).

Ahn and coworkers retrospectively examined the outcomes of 2054 kidney transplant recipients to evaluate the efficacy of lamivudine in HBsAg-positive recipients. There were 66 kidney transplant recipients with pretransplant HBsAg-positive serologic status. They found that the 10-year patient survival was greater in pretransplant HBsAg-negative recipients (*n* = 1988) than in HBsAg-positive patients on lamivudine (*n* = 27) and HBsAg-positive recipients without lamivudine (*n* = 39) [51] (Table 7). The 10-year death-censored graft survival was 79.7%, 69.6%, and 45.7% in the groups reported above, respectively. Pretransplant HBsAg-positive recipients who did not take lamivudine had significantly lower patient (*p* < 0.001) and graft (*p* < 0.001) survival than HBsAg-positive recipients on lamivudine.

Mo et al. [52] from Korea enrolled kidney transplant recipients who were propensity-matched in a 1:3 (HBV-positive: HBV-negative) fashion with regard to several covariates, such as age, gender, the year of transplant, ABO incompatibility, and cross-match positive status. The total number of HBsAg-positive patients recruited for the study was 77 (mean age, 47.1 ± 11.5 years, and 59 males). The authors identified three groups: HBsAg-positive patients on antiviral agents (*n* = 65), HBsAg-positive patients who did not receive antivirals (*n* = 12), and HBsAg-negative patients (*n* = 231). Significant differences occurred in patient survival among the three groups (*p* = 0.02). However, there was no difference in patient survival between HBsAg-positive patients who received antiviral agents and HBsAg-negative patients (NS) (Table 7). No significant difference in death-censored graft survival occurred among the three groups.

Yap and colleagues retrospectively evaluated the long-term outcomes of HBsAg-positive recipients who underwent kidney transplantation at Queen Mary Hospital during the years 1985–2008. A total of 63 HBsAg-positive and 63 HBsAg-negative patients were enrolled. Considering all HBsAg-positive transplanted patients during the study period of 23 years, antiviral treatment was associated with improved patient survival, with 20-year survival rates of 83% compared with 34%, respectively, in patients who did not receive antiviral treatment. Ten-year survival rates in the three groups are shown in Table 7. Liver-related complications accounted for 22.2% and 64.3% of deaths in patients who had and had not received antivirals (*p* = 0.036) [53].

In the study by Chan and coworkers [28], there were 12 HBsAg-positive transplant recipients who had undergone kidney transplantation between 1996 and 2000; 11 of them received lamivudine (pre-emptive approach) (first group). Lamivudine was initiated based on HBV DNA values in some (*n* = 5) recipients, whereas others (*n* = 6) also had abnormal liver biochemistry. The first group (*n* = 12) had similar survival to that of HBsAg-negative patients (*n* = 442) (Table 7).

A recent systematic review with a meta-analysis on the outcomes of kidney transplantation in patients with HBV infection was published recently [25]. According to their meta-regression, Thongprayoon and coworkers found a negative relationship between the death risk after a kidney transplant in HBsAg-positive patients and the year of publication (slope = −0.062, *p* = 0.001). This implies the improved management of chronic HBV after renal transplant with oral antiviral agents [25].

## 8. Kidney Transplants from Donors with Hepatitis B and Antiviral Agents

The availability of antiviral agents for HBV supports challenging treatment options such as kidney transplants from HBsAg-positive donors. Indeed, novel strategies to increase the donor supply and reduce the discrepancy between organs needed and organs available remain a priority. Based on the Kidney Disease Improving Global Outcomes (KDIGO) Clinical Practice Guideline on the Evaluation and Care of Living Kidney Donors, the transplantation of kidneys from HBsAg-positive donors is contraindicated for HBV- recipients, but may be considered for HBsAg-positive recipients or recipients with HBV protective immunity, with the informed consent of the recipient, possible antiviral HBV treatment of the recipient, and posttransplant monitoring [54]. Recent trials encourage transplant programmes with a large frequency of HBsAg-positive donors to allocate kidneys from HBsAg-positive donors to HBsAg-negative recipients [55].

## 9. Conclusions and Personal Views

Some drugs are now available for the treatment of HBV, and these proved to be effective and safe even in patients with advanced CKD who have undergone kidney transplantation. Recent evidence has been accumulated showing that antiviral drugs for chronic HBV improve patient and graft survival; however, in these studies, detailed clinical information (i.e., patient comorbidities) was lacking. Kidney transplant recipients need to receive immunosuppressive therapy throughout their lives, and consequently, antiviral agents should be continued indefinitely, as outcomes after the withdrawal of antiviral treatment in kidney transplant recipients with chronic HBV infection remain unknown. Antiviral therapy should be routinely proposed to HBV-positive kidney transplant recipients and candidates to avoid the deleterious hepatic and extra-hepatic effects of chronic and active HBV infections. Another limitation of this study is the limited evidence regarding novel strategies including the use of antivirals in HBsAg-negative recipients who receive kidney transplants from HBsAg-positive donors.

## Figures and Tables

**Figure 1 pathogens-12-00340-f001:**
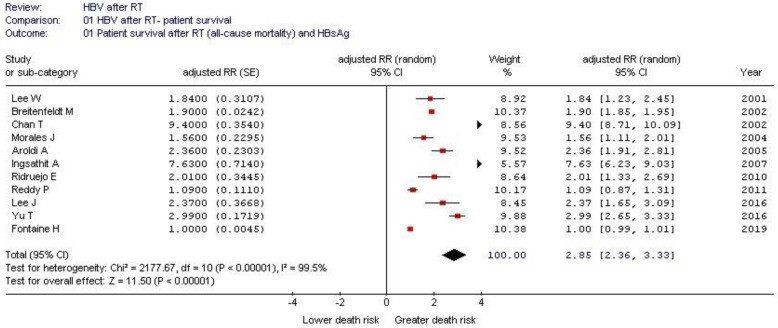
Forest plot: impact of HBV infection on all-cause mortality after kidney transplant [10,12,14,26,27,28,29,30,31,32,33].

**Figure 2 pathogens-12-00340-f002:**
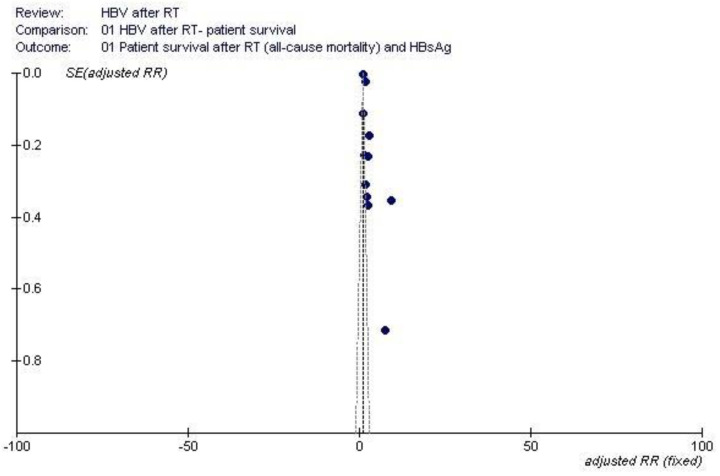
Funnel plot: HBV and patient survival after kidney transplant.

**Figure 3 pathogens-12-00340-f003:**
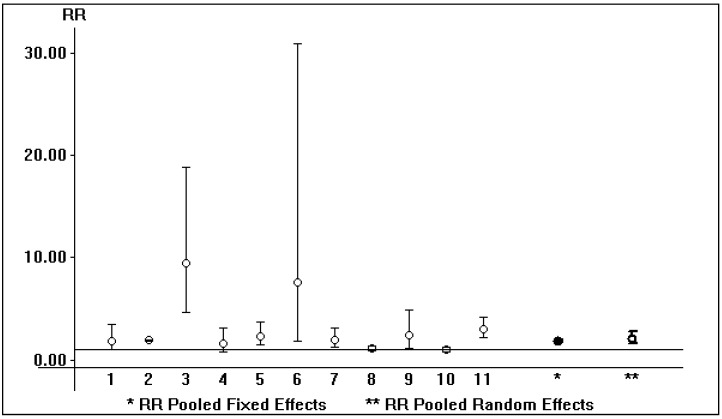
HBsAg-seropositive status and patient survival: aRR of all-cause death (according to fixed- and random-effects model). * RR pooled fixed effects; ** RR pooled random effects.

**Figure 4 pathogens-12-00340-f004:**
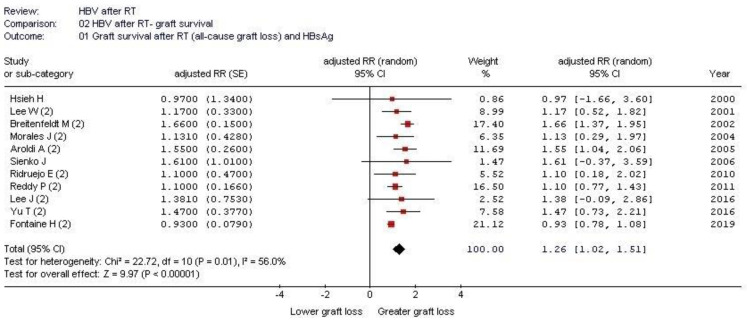
Forest plot: impact of HBV infection on all-cause graft loss after kidney transplant [10,12,14,26,27,29,30,32,33,34,35].

**Figure 5 pathogens-12-00340-f005:**
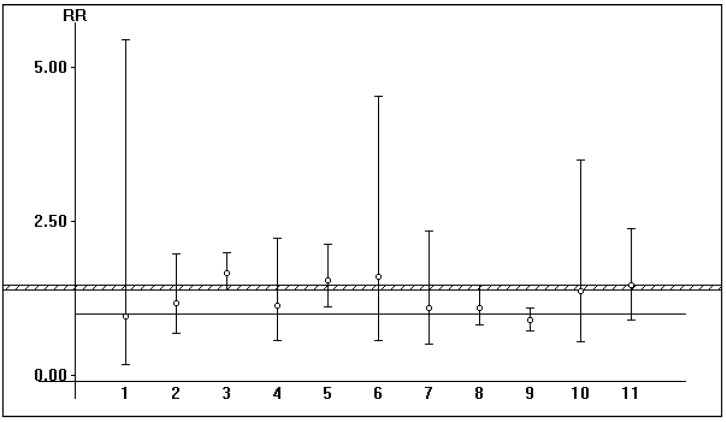
Odd Man Out: HBsAg-seropositive status and all-cause graft loss: aRR of graft loss and 95% CI (individual studies). Vertical bars indicate the 95 percent confidence intervals (95% CIs) for the relative risk of graft loss after kidney transplant (in each study) according to HBsAg-seropositive status. The shaded region indicates the portions of the RR of the graft loss axis included in the eleven study-specific 95% CIs.

**Table 1 pathogens-12-00340-t001:** Prevalence of HBsAg-seropositive status in advanced CKD worldwide.

Patients, *n* [Ref. Number]	HBsAg Rate, *n*	Country	Population	Reference Year
Raina D, et al. [6]	11.7% (7/60)	India	HD	2022
Jeele M, et al. [7]	7.3% (16/220)	Somalia	HD	2021
Ko K, et al. [8]	1.3% (19/1437)	Japan	HD	2020
Fontaine H, et al. [10]	1.8% (575/31,433)	France	RT	2019
Jadoul M, et al. [5]	2.3% (1898/82,449)	DOPPS	HD	2019
Meng C, et al. [11]	0.08% (7/973)	Portugal	RT	2018
Lee J, et al. [12]	4.5% (160/3482)	Korea	RT	2016
Garcia Agudo R, et al. [13]	1% (156/15,645)	Spain	HD	2016
Yu T, et al. [14]	9.3% (386/4133)	Taiwan	RT	2016
Bah A, et al. [15]	8.3% (48/579)	Guinea	CKD (including HD)	2015
Grenha V, et al. [16]	3.3% (76/2284)	Portugal	RT	2015
Wang C, et al. [9]	11.9% (707/5941)	China	HD	2010
Thanachartwet V, et al. [17]	6.5% (169/2585)	Thailand	HD	2007
Vladutiu D, et al. [18]	21.8% (23/108)	Romania	HD	2000

**Table 2 pathogens-12-00340-t002:** Baseline characteristics of studies included in the pooled analysis.

Author [Ref. Number]	Country	Reference Year	Study Design	Study Size	HBV Treatment
Lee WC, et al. [26]	Taiwan	2001	Retrospective cohort	477	NA
Breitenfeldt M, et al. [27]	Germany	2002	Retrospective cohort	927	NA
Chan T, et al. [28]	Hong Kong	2002	Prospective cohort	509	26/67 (39%)
Morales J, et al. [29]	Spain	2004	Multi-centre retrospective cohort	3365	NA
Aroldi A, et al. [30]	Italy	2005	Retrospective cohort	541	NA
Ingsathit A, et al. [31]	Thailand	2007	Retrospective cohort	346	NA
Ridruejo E, et al. [32]	Argentina	2010	Retrospective cohort	542	NA
Reddy P, et al. [33]	USA	2011	Population-based retrospective cohort	75,681	NA
Lee J, et al. [12]	Korea	2016	Multi-centre retrospective cohort	3482	129/160 (80.6%)
Yu TM, et al. [14]	Taiwan	2016	Population-based retrospective cohort	4133	232/386 (60%)
Fontaine H, et al. [10]	France	2019	Population-based prospective cohort	31,433	143/167 (85.6%)

**Table 3 pathogens-12-00340-t003:** Baseline characteristics of studies included in the pooled analysis.

Author [Ref. Number]	Diabetes, *n*	Mean Follow-Up after RT (mo)	HCV-Positive Patients after RT, *n*	Hypertension, *n*
Lee WC, et al. [26]	20 (4.2%)	72 ± 84	151 (31.6%)	327 (68.6%)
Breitenfeldt M, et al. [27]	NA	110.4 ± 52.8	130 (14%)	NA
Chan T, et al. [28]	NA	82 ± 58/NA	0	NA
Morales J, et al. [29]	407 (12%)	NA	513 (15%)	NA
Aroldi A, et al. [30]	NA	172.8 ± 67/168 ± 60	244 (45%)	NA
Ingsathit A, et al. [31]	3 (14%)/37 (11.4%)	44.4 (6–81)	22 (6.3%)	NA
Ridruejo E, et al. [32]	NA	76.8 ± 59.5	180 (33%)	NA
Reddy P, et al. [33]	385 (28%)/23,044 (31%)	NA	0	17,727 (23.4%)
Lee J, et al. [12]	580 (16.7%)	NA	55 (1.6%)	2856 (82%)
Yu T, et al. [14]	752 (18.2%)	NA	312 (7.5%)	2874 (6.9%)
Fontaine H, et al. [10]	2106 (6.7%)	NA	73 (0.002%)	11,510 (36.6%)

**Table 4 pathogens-12-00340-t004:** Baseline (and virological) characteristics of studies included in the pooled analysis.

Author [Ref. Number]	Age, Years	Males, *n*	HBsAg-Positive	HBeAg-Positive	HBV-DNA-Positive
Lee W, et al. [26]	38.6 ± 11.5	280 (58.7%)	62/477 (12.9%)	NA	NA
Breitenfeldt M, et al. [27]	40 ± 12/42 ± 13	595 (64.1%)	37/927 (3.9%)	11/37 (29.7%)	NA
Chan T, et al. [28]	NA	NA	67/509 (13.2%)	23/49 (46.9%)	NA
Morales J, et al. [29]	NA	NA	76/3365 (2.2%)	NA	NA
Aroldi A, et al. [30]	34 ± 11/31 ± 12	322 (59.5%)	77/541 (14.2%)	34/77 (44%)	43/65 (66.1%)
Ingsathit A, et al. [31]	NA	215 (62.1%)	23/346 (6.6%)	NA	NA
Ridruejo E, et al. [32]	42.0 ± 13	325 (60%)	23/542 (4.2%)	NA	NA
Reddy P, et al. [33]	NA	45,249 (59.8%)	1346/75,681 (1.8%)	NA	NA
Lee J, et al. [12]	40.6 ± 12.9	2084 (59.9%)	160/3482 (4.6%)	NA	NA
Yu TM, et al. [14]	NA	2140 (51.8%)	386/4133 (9.3%)	NA	NA
Fontaine H, et al. [10]	49.4	19,579 (62.3%)	575/31,433 (1.8%)	NA	146/167 (87.4%)

**Table 5 pathogens-12-00340-t005:** Stratified analysis: summary estimates for adjusted RR of all-cause mortality and HBsAg after kidney transplant.

	Study, *n*	Fixed-Effects aRR (95% CI)	Random-Effects aRR (95% CI)	R*_i_*	*p*-Value (by the *Q*-Test)
All studies	11	1.03 (1.03; 1.04)	2.85 (2.36; 3,33)	0.97	0.0001
Single-centre studies	8	1.92 (1.84; 2.01)	2.75 (1.82; 4.16)	0.98	0.0001
Population-based studies	3	1.27 (1.1; 1.47)	1.16 (0.82; 2.62)	0.94	0.0000
Recent studies	5	1.35 (1.18; 1.55)	1.67 (1.06; 2.61)	0.90	0.0001
Studies from Asia	5	3.22 (2.51; 4.13)	3.58 (2.04; 6.28)	0.77	0.0056
Studies from Europe	4	1.86 (1.78; 1.95)	1.61 (1.06; 2.43)	0.98	0.0000
Studies based on high HBV load	2	1.21 (0.98; 1.49)	1.51 (0.65; 3.48)	0.94	0.0001
Studies based on low HBV load	2	1.91 (1.83; 2.0)	4.06 (0.85; 19.4)	1.0	0.0001

**Table 6 pathogens-12-00340-t006:** Stratified analysis: summary estimates for adjusted RR of all-cause graft survival and HBsAg after kidney transplant.

	Study, *n*	Fixed-Effects aRR (95% CI)	Random-Effects aRR (95% CI)	R*_i_*	*p*-Value (by the *Q*-test)
All studies	11	1.13 (1.01; 1.25)	1.26 (1.02; 1.51)	0.60	0.01
Single-centre studies	6	1.56 (1.35; 1.81)	1.56 (1.35; 1.81)	0	0.74
Population-based studies	3	1.32 (0.95; 1.82)	1.32 (0.95; 1.82)	0	0.9
Recent studies	4	1.01 (0.87; 1.18)	1.06 (0.86; 1.31)	0.38	0.22
Studies from Asia	4	1.32 (0.95; 1.82)	1.32 (0.95; 1.82)	0	0.9
Studies from Europe	5	1.3 (1.15; 1.47)	1.31 (0.93; 1.85)	0.84	0.003
Studies based on high HBV load	2	1.05 (0.88; 1.24)	1.16 (0.68; 1.98)	0.9	0.005

**Table 7 pathogens-12-00340-t007:** Ten-year patient survival after kidney transplant according to HBsAg status and antiviral therapy for HBV.

Author [Ref. Number]	HBsAg-Negative pts (*n*)	HBsAg-Positive on Antiviral Therapy (*n*)	HBsAg-Positive without Antiviral Therapy (*n*)	Antiviral Agent	Country
Ahn H, et al. [51], 2007	88.2% (*n* = 1988)	85.3% (*n* = 27)	49.9% (*n* = 39)	LAM	South Korea
Yap D, et al. [53], 2010	100% (*n* = 63)	90% (*n* = 38)	55% (*n* = 25)	LAM	Hong Kong
Lee J, et al. [26], 2016	83.6% (*n* = 3267)	100% (*n* = 46)	NA	LAM, ADV, ETV	Korea
Mo H, et al. [52], 2021	98% (*n* = 231)	96% (*n* = 55)	78% (*n* = 12)	LAM, ADV, LDT, TDF, ETV	South Korea
Chan T, et al. [28], 2002	96% (*n* = 442)	100% (*n* = 12)	71% (*n* = 55)	LAM	Hong Kong

ADV = adefovir dipivoxil; ETV = entecavir; LAM = lamivudine; LDT = telbivudine; TDF = tenofovir disiproxil.

## Data Availability

Not applicable.

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
