# Peer review of "The Impact of Antiviral Treatment of Hepatitis B Virus after Kidney Transplant and the Latest Insights"

_pathogens, 2023, doi:10.3390/pathogens12020340_

Round 1
Reviewer 1 Report (Previous Reviewer 1)
The authors have modified the manuscript and now it is acceptable for the publication.
Author Response
I have revised the manuscript according to the comments of Reviewers and Editor
Reviewer 2 Report (New Reviewer)
In their review, Fabrizi and coauthors describe the results of the performed analysis of published data on outcomes of HBV infection and approaches to manage this infection in patients with chronic kidney disease and those who undergone renal transplantation.
The analysis seems to be comprehensive, addresses the actual issues in HBV-related morbidity and mortality in patients with chronic kidney disease, and highlights the gaps existing in current knowledge on the topic. However, there are some minor comments that should be addressed.
1. Please provide the explanation for CKD in Abstract to make it stand-alone.
2. Why only entecavir, but not NAs in general is indicated as a keyword?
3. Please add to Introduction the brief review of current recommendations for management of HBV infection and chronic hepatitis B in patients with chronic kidney disease and renal transplantation (EASL, AASL).
4. Section 3 “…and other countries including China, Russia and Turkey”. Perhaps, not Russia but Romania, as indicated in Table 1.
5. It is advisable to place Section 7 before Section 6, i.e., to provide the description of treatment options for patients with kidney disease before the data on the impact of such treatment.
6. The term “activity” used in Section 6 title is not appropriate. It is rather therapy effects or outcomes than “activity”.
7. Table 7 – Please add data on NAs used.
8. Section 8 – Please provide a brief description of the recommended treatment strategies and donor/recipient characteristics mentioned in this section.
Author Response
Milano February 12, 2023
Prof Dr Lawrence S Young
Professor of Molecular Oncology
Director, Division of Cancer Studies
Warwick Medical School,
University of Warwick
Coventry, UK
Dear Prof L Young:
Please find enclosed the revised version of my manuscript (manuscript ID- 2214141) entitled:
‘The impact of antiviral treatment of hepatitis B virus after kidney transplant and the last insights’ for your evaluation for publication in the journal Pathogens (narrative review).
The revised version of the manuscript has the following changes:
- I have highlighted (yellow) the changes;
- I have provided the explanation for CKD in Abstract (Reviewer #2, point #1);
- I have indicated NAs, instead of ‘entecavir’ as a key-word (Reviewer #2, point #2);
- I have added some sentences (Section Introduction; page #5) regarding ‘current recommendations for management of HBV infection and chronic hepatitis B in patients with chronic kidney disease and renal transplantation (EASL, AASLD) (Reviewer #2, point #3). Also, I have added another Ref (Ref #54);
- I have changed ‘ ..and other countries including China, Russia and Turkey’ (page #8) (Reviewer #2, point #4);
- I have placed Section 7 before 6 (page #13) (Reviewer #2, point #5);
- I have deleted the term ‘activity’ (Reviewer #2, point #6);
- I have added data on NAs in Table 7 (Reviewer #2, point #7);
- I have provided a brief description of the recommended treatment strategies (page #18) (Reviewer #2, point #8);
- I have changed some Figure legends (Reviewer #3) (page #34)
I look forward to hearing from you soon.
Thank you for your kind attention.
Best wishes,
Fabrizio Fabrizi, MD
Professor of Medicine and Associate Director
Division of Nephrology, Dialysis and Kidney Transplant
Foundation IRCCS Cà Granda, Ospedale Maggiore Policlinico
Milano, Italy
Reviewer 3 Report (New Reviewer)
Manuscript ID: pathogens-2214141
Type of manuscript: Review
Title: The Impact of Antiviral Treatment of Hepatitis B Virus after Kidney
Transplant and the Last Insights
Authors: Fabrizio Fabrizi et al.
This was a comprehensive and educational review. However, the selected reports were a little old, in particular anti-HBV drugs.
European Association for the Study of the Liver recommends that all HBsAg-positive renal transplant recipients should receive ETV or TAF as prophylaxis or treatment (Evidence level II-2, grade of recommendation 1). Authors should address the management by hepatologists in Discussion.
Minor;
Figure legends were not enough. It was difficult to understand figures.
Authors should revise them more closely and more politely.
For examples, there was no information about the number 1-11 in figure 3 and 4. What meant two lines; the simple horizontal line and the striped line?
Unsuitable new line and unnecessary space should be revised in the text.
Author Response
Milano February 12, 2023
Prof Dr Lawrence S Young
Professor of Molecular Oncology
Director, Division of Cancer Studies
Warwick Medical School,
University of Warwick
Coventry, UK
Dear Prof L Young:
Please find enclosed the revised version of my manuscript (manuscript ID- 2214141) entitled:
‘The impact of antiviral treatment of hepatitis B virus after kidney transplant and the last insights’ for your evaluation for publication in the journal Pathogens (narrative review).
The revised version of the manuscript has the following changes:
- I have highlighted (yellow) the changes;
- I have provided the explanation for CKD in Abstract (Reviewer #2, point #1);
- I have indicated NAs, instead of ‘entecavir’ as a key-word (Reviewer #2, point #2);
- I have added some sentences (Section Introduction; page #5) regarding ‘current recommendations for management of HBV infection and chronic hepatitis B in patients with chronic kidney disease and renal transplantation (EASL, AASLD) (Reviewer #2, point #3). Also, I have added another Ref (Ref #54);
- I have changed ‘ ..and other countries including China, Russia and Turkey’ (page #8) (Reviewer #2, point #4);
- I have placed Section 7 before 6 (page #13) (Reviewer #2, point #5);
- I have deleted the term ‘activity’ (Reviewer #2, point #6);
- I have added data on NAs in Table 7 (Reviewer #2, point #7);
- I have provided a brief description of the recommended treatment strategies (page #18) (Reviewer #2, point #8);
- I have changed some Figure legends (Reviewer #3) (page #34)
I look forward to hearing from you soon.
Thank you for your kind attention.
Best wishes,
Fabrizio Fabrizi, MD
Professor of Medicine and Associate Director
Division of Nephrology, Dialysis and Kidney Transplant
Foundation IRCCS Cà Granda, Ospedale Maggiore Policlinico
Milano, Italy
Round 2
Reviewer 3 Report (New Reviewer)
Thank you for replies.
This manuscript is a resubmission of an earlier submission. The following is a list of the peer review reports and author responses from that submission.
Round 1
Reviewer 1 Report
The authors have made an interesting attempt at “The Impact of Antiviral Treatment of Hepatitis B Virus after Kidney Transplant and the Last Insights.” The manuscript is interesting; however, the authors need to justify the scientific writing manuscript. Some of the general comments are provided below:
1. The authors should mention the inclusion and exclusion criteria of studies from January 1990 to June 2022 in a more organized form.
2. Retrieved studies about the activity of antiviral treatments after kidney transplants were not adequately explained.
3. Were there patients with comorbidities in five studies that formally evaluated the effects of antiviral treatment towards HBV and reported consistent improvement of patient and graft survival with the adoption of antiviral therapies towards HBV?
4. Highly effective direct-acting antivirals against the Hepatitis C virus (HCV) have created an opportunity to transplant organs from HCV-positive individuals into HCV-negative recipients since de novo infection can be routinely cured. Are there any studies showing the use of organs from HBV-positive patients for kidney transplants?
5. Is there any impact of different genotypes on graft survival and antiviral therapies toward HBV?
6. What are the limitations of this study?
Reviewer 2 Report
The review resembles more to a reanalysis of previous studies than a real review. The authors should reconsider rewriting it by pointing out the most important findings of these studies and disucss it.